# Examining Relationships among Chinese Preschool Children’s Meeting 24-Hour Movement Guidelines and Fundamental Movement Skills

**DOI:** 10.3390/jcm11195623

**Published:** 2022-09-24

**Authors:** Fang Li, Long Yin, Mingyun Sun, Zan Gao

**Affiliations:** 1School of Physical Education, Hengyang Normal University, Hengyang 421002, China; 2School of Mathematics and Statistics, Hengyang Normal University, Hengyang 421002, China; 3School of Kinesiology, University of Minnesota-Twin Cities, Minneapolis, MN 55455, USA

**Keywords:** locomotor skills, object control skills, physical activity, sleep duration, screen time

## Abstract

Background: Few studies have investigated associations between meeting 24-Hour Movement Guidelines and preschool children’s fundamental movement skills (FMS). This study aimed to investigate the associations between Chinese preschoolers meeting various combinations of the Guidelines and FMS, as well as examining gender differences across those outcomes. Methods: A total of 322 preschool children (181 boys and 141 girls) aged 3–6 years old were recruited from five early-childhood education and care services in China. Children’s 7-day physical activity (PA) was assessed using accelerometry, and screen time (ST) and sleep duration were reported by their parents. FMS were assessed by *Test of Gross Motor Development 3*. Descriptive statistics and gender differences in children’s meeting the Guidelines and FMS were calculated using *t*-tests for continuous variables and Chi-square tests for categorical variables. Results: Most preschoolers met the sleep and ST guidelines, but only 22.3% met the PA guidelines; 4% of preschoolers did not meet any of the recommendations, while 9.3% met all three recommendations. Boys reported significantly higher compliance with all combined recommendations. There were positive associations between meeting the PA guidelines and both locomotor (β = 0.49; *p* < 0.001) and object control skills (β = 0.21; *p* < 0.001). A significantly positive relationship was identified between meeting PA + sleep guidelines and locomotor skills (β = −0.16; *p* = 0.02). A significantly negative association was observed between preschoolers’ meeting sleep + ST guidelines and locomotor skills (β = 0.31; *p* = 0.001). Conclusion: Few preschoolers met all three movement behavior guidelines. The findings suggest that PA levels, especially MVPA, are important for the promotion of FMS. Meanwhile, the impacts of ST and sleep on motor development cannot be neglected.

## 1. Introduction

As the foundation of complex, task-specific movements, and fundamental movement skills (FMS) are the synthesis of movement techniques and physical competence [1,2], and also the basic skills that form the prerequisites for individuals to engage in sports and physical activity (PA) [3]. There are three aspects of FMS: locomotor skills (e.g., running and hopping), object control skills (e.g., catching and throwing), and stability (e.g., balancing and twisting) [4]. It has been evident that children’s PA levels are positively related to their FMS [5,6,7,8], with an especially close positive association between moderate-to-vigorous physical activity (MVPA) and object control skills [9]. Interestingly, findings from a recent systematic review reported no significant relationship between children’s light PA (LPA) and FMS [10]. Although evidence for the relationship between locomotor skills and MVPA [11,12] has been uncertain, it is still recommended that young children should spend at least 180 min on PA, including at least 60 min of MVPA per day [13].

Research evidence indicated that FMS in preschoolers was negatively related to their sedentary behavior (SB) [12], which mainly comprised screen time (ST). Therefore, the World Health Organization’s 24-Hour Movement Guidelines recommended that 3–4-year-old preschoolers should have limited recreational ST of no more than one hour per day, and 5-year-olds should have no more than two hours per day [13,14,15]. It is also known that adequate sleep plays a fundamental role in the memory consolidation process and in the acquisition and retention of information [16]; thus, researchers postulated that inadequate sleep is negatively correlated with FMS acquisition [17]. According to the World Health Organization Guidelines, those 3–4-year-olds should have from 10 to 13 h of good-quality sleep per day, and 5-year-olds should have from 9 to 11 h per day [13,14,15].

Although a number of studies explored the associations between children meeting single movement behavior recommendations (i.e., PA, SB, and Sleep) and their FMS [12], some scholars called for the need to not only consider the impact of children’s MVPA on their health and FMS, but also to comprehensively consider the joint effects of LPA, SB, sleep and other behaviors on the promotion of health and FMS [18,19]. The “Activity Balance Model” and “Time-Use Epidemiology” proposed by Pedisic posited that sleep duration, SB, standing, LPA, and MVPA should all be considered an Integration [20,21]. Researchers further suggested that children’s meeting all three 24-Hour Movement Guidelines (i.e., PA, SB, Sleep is related to more physical and FMS benefits than meeting just one or none of the recommendations [10,22,23,24]. The research literature suggested that, in general, boys demonstrated greater PA or MVPA than girls in their daily life [7,25,26,27]. Researchers also indicated that, in childhood and adolescence, there is no difference in locomotor skills between genders, but boys were more proficient in object control skills than girls [28]. For balance skills, empirical evidence suggested that girls performed better than boys, yet PA participation was not significantly associated with children’s balance skills [29]. The underlying mechanisms of the development of balance skills might involve other factors and deserves further investigation.

A recent study revealed that only 5.12% of Chinese children and adolescents met all three 24-Hour Movement Guidelines [30]. To our knowledge, researchers have not explained the compositional behaviors of PA, ST, and sleep throughout the day, or their relationships with FMS in a sample of Chinese preschool children. Thus, this study aimed to investigate the association between Chinese preschoolers’ meeting various potential combinations of the 24-Hour Movement Guidelines and FMS, as well as examining gender differences in adherence to the 24-Hour Movement Guidelines.

## 2. Materials and Methods

### 2.1. Design and Participants

For this study, preschoolers aged 3–6 years of both genders from Heng Yang, a middle-sized city in the south–central region of Hunan Province in China, were recruited. All preschool children were registered in early childhood education and care services (ECECs) licensed by the Hengyang Municipal Bureau of Education. A total of 426 preschoolers with no chronic disease from five ECECs were invited to participate in this study. Parents were notified about the study via letter and provided written informed consent if they allowed their children to attend this study. Thirty parents did not provide consent for their children, and 12 children did not attend the preschool during assessment days. The sample comprised 384 preschoolers, with 62 not providing complete data concerning PA, FMS, and/or sleep duration/ST. Therefore, the final sample included 322 preschool children, with 181 boys and 141 girls (M = 4.69 years old, SD = 0.048). The study protocol was approved by the Institutional Review Board of Hengyang Normal University in 2021 (NO.2021003).

### 2.2. Measures

#### 2.2.1. PA

Preschoolers’ PA was objectively measured using waist-worn accelerometry (Actigraph, model wGT3X-BT, USA), which has been established as a valid instrument to measure PA in this target population [28]. Kindergarten teachers and parents were given oral and written guidelines for the proper use of accelerometers, including how and when to wear them. Children were asked to wear the accelerometer on the right hip for seven consecutive days, except for water-based activities and sleeping. The accelerometry initialization and data were analyzed by the ActiLife software (ActiGraph Corps., Pensacola, FL, USA; Version 6.13.3). We adopted the cut-points of PA intensity from Butte (2014) in this study. The specific measurement parameters of the accelerometers are shown in Table 1. The total PA was calculated as LPA + MPA + VPA, and the MVPA was calculated as MPA + VPA.

#### 2.2.2. Screen Time and Sleep Duration

Children’s ST and sleep duration were reported by parents via questionnaire, including daytime sleep hours and nighttime sleep hours. ST included TV time, computer time, smartphone time, and video-games time. Questions such as “how many hours does your child spend watching TV, using the computer, using a smart phone, or playing videogames on weekdays?” were asked to the parents. Subsequently, average ST was calculated as: ([ST on weekdays × 5] + [ST on weekend × 2])/7.

The sleep duration was reported by parents through a validated questionnaire [33], including the bedtime and wakeup time, which were counted separately for weekdays and weekends, during the day and night. Preschooler’s daytime sleep duration was calculated as: (([wake up time on weekdays–bedtime on weekdays] × 5) + ([wake up time on weekends–bedtime on weekends] × 2))/7. The same procedure was used to calculate nighttime sleep. The daily sleep duration was calculated as daytime sleep time + nighttime sleep time.

#### 2.2.3. FMS

The “*Test of Gross Motor Development 3*” (TGMD-3) was used to assess preschoolers’ FMS [34]. This includes two subtests: locomotor and object control skills. The locomotor skills test contains six tasks: run, skip, slide, gallop, hop, and horizontal jump. The object control skills test contains seven tasks: overhand throw, underhand throw, catch, dribble, kick, one-hand strike, and two-hand strike. Each child performed the testing tasks twice in a safe environment under the instruction of a trained instructor. The whole process was recorded with a camera. Then, two trained research assistants scored each child’s skill performance based upon the recorded videos. If there was an apparent difference between the two scores of each skill, a third observer would intervene and reconcile the scoring. Each skill was evaluated by 3–5 specific criteria. If a child performed correctly on a skill, we would assign score 1 to him or her; if a child could not perform correctly, we would assign score 0 to him or her. Skill scores were then summed to a raw skill score for the subscale scores (i.e., locomotor skills, and object control skills). Note that the maximum raw score for locomotor skills was 46 points, and 54 points was the maximum for the object control skills. That is, the maximum score for overall motor skills was 100 points, with a higher score reflecting better motor skill performance.

Approximately 10% of the videos were randomly selected and analyzed by the researchers twice, with an interval between the two evaluations of one week, to determine the intra-group correlation coefficient (ICC). In this study, high degrees of consistency were observed in FMS testing scores: locomotor skills score ICC = 0.92 (95% confidence interval: 0.62–0.97), object control skills score ICC = 0.95 (95% confidence interval: 0.89–0.98), FMS score ICC = 0.94 (95% confidence interval: 0.83–0.98).

#### 2.2.4. Demographic Status

Demographic information about the preschool children, such as age, gender, height and weight, were reported by the kindergarten physician. Family socio-economic status (SES) included parents’ education level, parents’ occupation and family monthly income. Parents were asked to fill out a questionnaire on SES, adopted from an established research [35]. Then, we calculated the SES index using principal component analysis. Next, children’s SES was classified into three grades, namely, low, medium, and high SES, according to the method of standard deviation.

### 2.3. Statistical Analysis

All data were sorted and cleaned with Excel (Version 2019; Microsoft Cooperation, Redmond, WA, USA). Participants were classified as adhering or not adhering to the individual 24-Hour Movement Guidelines (i.e., not meeting any recommendations, meeting PA only, meeting ST only, and meeting SL only) and in combination (i.e., meeting PA + ST only, meeting PA + SL only, meeting ST + SL only, and meeting all three recommendations).

Descriptive statistics, including mean, standard division (SD), and percentages were performed. Gender differences in 24-Hour Movement Guidelines and FMS were calculated using *t*-tests for continuous variables and Chi-square tests for categorical variables in SPSS (Version 26.0; IBM, Armonk, NY, USA).

The confirmatory factor analysis was used to evaluate the TGMD-3 measurement model. Then, a structural equation modelling analysis was performed via Mplus 8.0 (Muthén and Muthén, 2017) to evaluate the relationship between preschoolers’ FMS and their meeting the 24-Hour Movement Guidelines. The structural equation modelling was adjusted for age, sex, SES and BMI. All statistical significance was set a priori at *p* < 0.05.

## 3. Results

### 3.1. Descriptive Analysis

The preschooler’s descriptive characteristics are shown in Table 2. On average, preschoolers exceeded the total PA-time guideline but did not meet the MVPA-time guidelines, and fell within sleep and ST guideline ranges. In detail, boys spent an average of 267 min per day on PA with 52 min for MVPA per day; 10.64 h of sleep per day; and 1.71 h per day on screen. Girls spent an average of 247 min per day on PA with 44 min for MVPA per day; 10.86 h of sleep per day; and 1.71 h per day on screen. Boys spent more time on PA than girls, but girls spent more time on sleep than boys. There were no significant differences in ST between boys and girls. In addition, boys performed better on slide, dribble, handpass, kick, upthrow, and roll, while girls performed better on leap and catch.

The proportions of preschoolers meeting 24-Hour Movement Guidelines separately and in all possible combinations are shown in Figure 1 and Table 3. Of the sample, most preschoolers met the sleep (90.7%) and ST (59.4%) guidelines, but only 22.3% met the PA guidelines. Only 4% did not meet any of the recommendations, while 9.3% met all three recommendations. Compared to girls, boys reported significantly greater compliance with all three recommendations (14.4% vs. 2.8%), the combination of PA and sleep recommendations (25.97% vs. 7.8%), the combination of PA and ST recommendations (16.02% vs. 2.84%), and the combination of ST and sleep recommendations (50.8% vs. 46.8%). There were significant gender differences in meeting PA, PA + ST, PA + sleep and all three 24-Hour Movement Guidelines. There were no significant gender differences in meeting ST and the combination of sleep + ST recommendations.

### 3.2. Associations among Meeting the Guidelines and FMS

The original structural model of the TGMD-3 did not present adequate adjustment indexes in the Root Mean Square Error of Approximation (RMSEA = 0.04 [0.03–0.06]), Comparative Fit Index (CFI = 0.92), and Tucker–Lewis Index (TLI = 0.90), including 13 items: 6 locomotor and 7 object control skills. The factorial run and dribble loadings were low (<0.2), so we tested a second structural model without run and dribble. The second structural model presented adequate adjustment indexes in RMSEA (0.05) [0.03–0.06], CFI (0.95), and TLI (0.93).

As Table 4 and Figure 2 show, children meeting the PA guidelines was not only positively related with locomotor skills (β = 0.49; *p* < 0.001) but was also associated with object control skills (β = 0.21; *p* < 0.001). There were no other positive associations among preschoolers meeting other single guidelines and FMS. In terms of the relationship between preschoolers meeting the combination of 24-Hour Movement Guidelines and FMS, we found significantly positive associations between PA + sleep recommendations and locomotor skills (β = 0.31; *p* = 0.001). There were negative associations between meeting ST + sleep guidelines and locomotor skills (β = −0.16; *p* = 0.02). Meeting the single 24-Hour Movement Guidelines explained 61% and 48% of the variance in locomotor skills and object control skills, respectively. Meeting the combinations of 24-Hour Movement Guidelines accounted for 62% and 52% of the variance in locomotor skills and object control skills, respectively.

## 4. Discussion

### 4.1. Descriptive Findings of Meeting the Guidelines and Gender Differences

This study reported the proportion of preschoolers meeting the 24-Hour Movement Behavior Guidelines in Hengyang City in China. Few Chinese preschoolers adhered to all three movement behavior guidelines (9.3%), and only 22.3% of the preschoolers adhered to the PA recommendations. However, most preschoolers met the sleep guidelines (90.7%), and more than half of the preschoolers adhered to the ST recommendations (50.7%). The findings echo the results from a preschool study (n = 539) in Canada, where only 5% of preschool children met the overall guidelines, with 19.3%, 50.5%, and 83.1% meeting the PA, ST, and sleep recommendations, respectively. However, another study of Chinese kindergarten children in Beijing had a different result, where 65.4% of children met PA guidelines, 88.2% met ST guidelines, 29.5% met sleep guidelines, and 15% met all three guidelines [36]. A possible explanation for such differences is the use of accelerometer data collection (e.g., different cut points and different wearing-time criteria). For example, our study used cut-off points of 240/15 s for LPA and 2120/15 s for MVPA, whereas Guan’s study used the cut-off point of 26/15 s for LPA and 420/15 s for MVPA [36]. In our study, the participants were asked to wear the accelerometer for seven days, and we selected valid data for 2 weekdays and 1 weekend day. Interestingly, Guan et al. asked participants to wear the accelerometer for three days and then only selected valid data for only one day, which made them unable to differentiate between weekdays and weekend days [36]. The researchers suggested that three days is acceptable to track preschool children’s PA [37].

Though boys reported higher percentages of compliance with all single recommendations and combinations of recommendations from the Guidelines compared to girls, there were gender differences in terms of meeting PA, PA + ST, PA + sleep and all three 24-Hour Movement Guidelines. This result is similar to the study of Spanish adolescents [38]. As boys spend more time on PA, more exercise plans should be carried out to promote physical activity in girls. This finding is consistent with those from two studies in China [26,27]. A meta-analysis also confirmed that boys spent more time engaged in PA [25]. The gender difference may be related familial, social, and environmental factors. During childhood, boys may prefer team sports and outdoor games, whereas girls tend to choose dance and rhythmic activities, which tend to be less active. However, in terms of sleep duration, ST and combination of sleep + ST, there are no gender differences; most girls (90%) and boys (91.2%) met the sleep recommendations, more than half of the girls (51.8%) and boys (54.7%) followed the ST guidelines, and the percentage of compliance with the combination of sleep + ST was similar for girls (46.8) and boys (50.8). This result is significantly better than that found in a recent study by Tapia-Serrano [38], who reported that only 10.8% of boys and 14.6% of girls met the sleep + ST recommendation. The reason for this might be the cultural and traditional differences in different countries.

### 4.2. Relationships among Children’s Meeting the Guidelines and Their FMS

The primary aim of this study was to examine the correlation between preschoolers meeting 24-Hour Movement Guidelines and FMS (including locomotion and object control skills). First, preschoolers meeting the PA guideline was not only positively associated with locomotor skills, but also with object control skills. That is, the preschoolers who met the PA recommendations presented better locomotion and object control skills, which is in line with the findings from many other studies [10,39,40,41,42,43]. As the foundation for future movement, FMS can support and maximize opportunities for children’s participation in PA [44,45], and is the pathway to a lifetime of engaging in PA [44]. Conversely, the development of FMS needs time to be completed; thus, sufficient PA time could positively predict FMS [45,46,47,48]. It is evident that the more time is spent on PA, the more opportunities one will receive to promote one’s neuromotor development, which, in turn, will promote one’s FMS [40]. Therefore, there is a mutual association between PA and FMS [40]. However, there is no significant association between FMS and meeting the sleep guidelines or ST guidelines in the present study, similarly to the results found in Kracht’s study [43].

Most studies have only investigated the associations between individuals’ single movement behavior, especially MVPA, and health outcomes. Movement behavior is a continuum. Ignoring any particular behavior may have different effects on health, so there are increasing studies focusing on meeting the 24-Hour Movement Guidelines in relation to health. Many researchers found evidence supporting the idea that there are associations between meeting the Guidelines and health in different populations [30,49,50,51,52,53,54]. The studies exploring meeting the Guidelines and FMS have also attracted the attention of many scholars due to the important role that FMS plays in promoting health [10,43]. In addition, it is imperative to implement innovative and effective PA programs to enhance FMS and promote MVPA in the pediatric population [7,55,56], with the goal of helping them to develop and maintain a physically active lifestyle.

Second, there was a significantly positive association between meeting PA + sleep recommendations and locomotor skills. In other words, preschoolers who met PA and sleep recommendations demonstrated better locomotion skills in this study. It is widely accepted that a good quality of sleep is beneficial to health and that a shorter sleep duration will lead to poorer health outcomes [57]. Sleep is “the price we pay for plasticity,” as, during sleep, the neuroplasticity process occurs, which is essential for memory consolidation. Adequate sleep provides for the cost of the higher energy consumption required by the human body [58]; therefore, sleep is suggested to be an important component in the acquisition and development of skills. However, too long a duration of sleep will reduce the PA time; thus, adherence to the sleep recommendations is suggested to benefit FMS and health. It is evident that PA can predict locomotor skills and object control skills [2,59,60]. As object control skills are more complex and more difficult to develop than locomotor skills, they require greater cognitive demands and need more time to develop. Consequently, a positive association between meeting the combined recommendations of PA and sleep and locomotor skills, but not object control skills, was found in our study.

Third, there was a negative association between meeting ST + sleep recommendations and locomotor skills, and there was a negative association between meeting the combined recommendations of sleep and ST and locomotor skills. ST is a common sedentary activity at the preschool stage. A day is a constant 24-Hour; the more time one spends on a particular behavior, the less time is spent on other behaviors, so higher amounts of ST might decrease the time children spend engaging in PA [61]. This would lead to lower FMS scores in preschoolers [12,62]. In our study, only 50.7% of preschoolers met the ST recommendations, while most preschoolers met the sleep recommendations. Martin’s study of preschoolers in Brazil pointed to a positive association between meeting the combination of ST and sleep duration recommendations and locomotor skills [10], which is contrary to our view. There may be several reasons for this: first, the preschooler’s ST was only reported by their parent, so times where the preschoolers were on screen unsupervised would be missed; second, the proportion of adherence to the combinations of screen and sleep recommendation was higher than the proportion of adherence to the PA recommendation in our study, which means that children who adhered to the combinations of screen and sleep recommendation may have not adhered to the PA recommendation. However, while PA, especially MVPA, is still a strong predictor of locomotor skills [10], the influence of PA on locomotor skills is not enough to offset the negative impacts of sleep and ST, so the impact of meeting the three (PA + sleep + ST) recommendations on locomotor skills is not significant. The other reason for this may be the cultural differences between China and western countries, which leads to differences in the content taught for physical education in kindergarten.

### 4.3. Study Strengths and Limitations

This is the first study to explore the associations between Chinese preschoolers’ adherence to a single movement behavior or a combination of movement behavior and FMS. The strengths lie in its decent sample size, objective assessment of children’s PA using the accelerometry and FMS via TGMD-3, and advanced statistical techniques to answer the research questions. However, several limitations to our study should be noted. First, preschoolers’ sleep duration and ST were estimated through a report from their parents, which may have led to measurement bias. Future research should consider using objectively measured instruments. Second, the cross-sectional design can only explain the associations between compliance with movement behavior and FMS in preschoolers, but not the causal relationship. Future studies may use a factorial experimental design to evaluate the cause-and-effect relationships. Additionally, this study was conducted in a middle–large city in the south–central region of China; thus, the research findings cannot be generalized to other populations. Finally, we encountered some difficulties when using the accelerometers to objectively measure sleep duration in young children in the present study. A major challenge is that some children thought that wearing the ActiGragh accelerometers on the right hip during sleep would cause discomfort [63]; thus, we recorded the sleep duration via questionnaire. Researchers may use objective instruments to record preschool children’s sleep duration in the future.

## 5. Conclusions

It is evident that the quantity and nature of children’s PA, ST, and sleep has gone through some changes in pandemic era [64]. Therefore, it is critical to explore preschoolers’ movement behaviors and determine whether they met the guidelines. In our study, few preschoolers adhered to the PA and ST guidelines, but most preschoolers met the sleep guidelines, and only 9.3% met all three recommendations. Boys showed significantly greater compliance with all combined recommendations compared to girls. Our data indicated that meeting the PA recommendation was positively associated with locomotor and object control skills in preschooler children. In addition, combined compliance with PA and sleep recommendations was positively associated with locomotor skills, but the combination of meeting sleep and ST guidelines was negatively associated with locomotor skills. Although some differences were reported concerning the associations between adhering to single or combinational guidelines and FMS, we posit that adhering to all three movement behavior recommendations will benefit FMS the most.

## Figures and Tables

**Figure 1 jcm-11-05623-f001:**
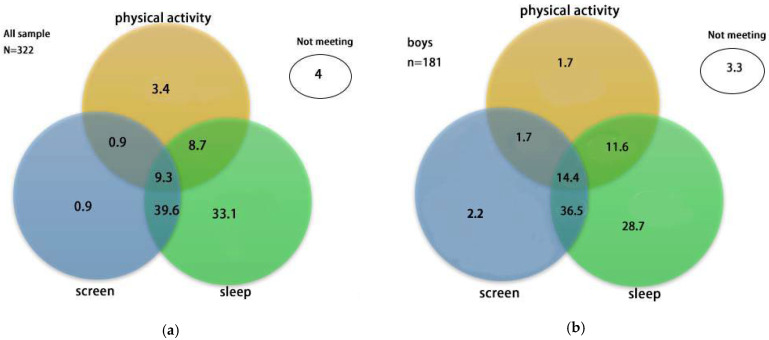
Venn diagram showing the proportion (%) of participants meeting 24-Hour Movement Guidelines separately and in all possible combinations. The sum of each circle is equivalent to the % meeting each individual recommendation (i.e., 22.3% for physical activity, 90.7% for sleep, 50.7% for ST, 4% for non-compliance in the full study sample). (**a**) Venn diagram for all the sample. (**b**) Venn diagram for boys. (**c**) Venn diagram for girls.

**Figure 2 jcm-11-05623-f002:**
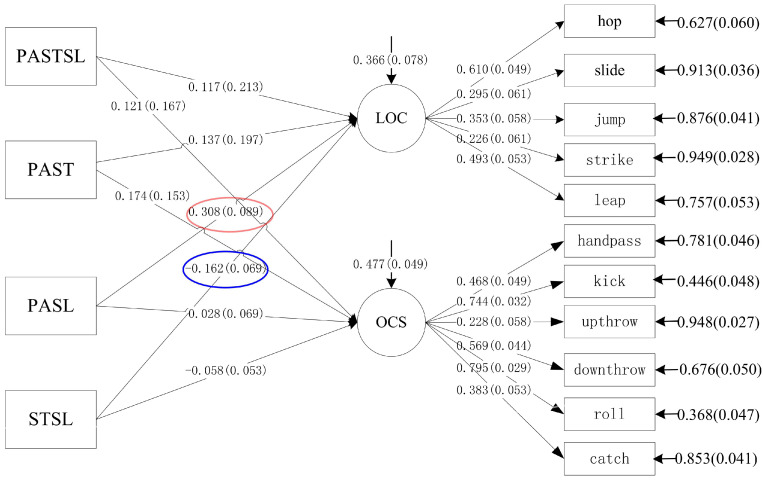
Structural model of the association of the combinations of 24-Hour Movement Behavior Guidelines that were met and fundamental movement skills in preschoolers. (PASTSL = physical activity + screen time + sleep; PAST = physical activity + screen time; PASL = physical activity + sleep time; STSL = screen time + sleep time; LOC = locomotor skills; OCS = object control skills). Colored with red line indicates significant positive relationship; Colored with blue line indicates significant negative relationship.

**Table 1 jcm-11-05623-t001:** Physical Activity Measurement Parameter Settings List of GT3X-BT.

Num	Parameter Content	Parameter Setting
1	Testing instrument	Actigraph WGT3X-BT
2	Sampling interval	15S
3	No definition is worn	Choi (2011) [31]
4	Valid data for every day	≥480 min
5	Valid data for wear day	At least 3 days (2 weekdays + 1 weekend)
6	Cut-points	Butte (2014) [32]
	SB	0–239
	LPA	240–2119
	MPA	2120–4449
	VPA	4450 and above

**Table 2 jcm-11-05623-t002:** Sample Characteristics and Differences by Gender.

Variables	Boys M (SD)	Girls M (SD)	*p*-Value
Age	4.74 (0.89)	4.63 (0.83)	0.06 *
hop	4.28 (2.23)	4.18 (2.20)	0.69
run	4.77 (1.99)	4.84 (2.17)	0.77
slide	4.87 (2.19)	4.26 (2.29)	0.02 *
jump	4.04 (2.05)	4.23 (2.01)	0.41
strike	4.66 (1.89)	4.89 (1.84)	0.26
leap	3.50 (2.18)	3.90 (1.97)	0.08 *
dribble	5.11 (1.78)	4.59 (1.78)	0.01 **
handpass	4.87 (2.28)	3.71 (2.37)	0.00 **
kick	2.48 (2.19)	1.72 (1.65)	0.00 **
upthrow	4.37 (2.33)	3.48 (2.12)	0.00 **
downthrow	4.72 (1.94)	4.47 (1.68)	0.21
roll	2.76 (2.17)	2.14 (1.96)	0.01 *
catch	3.77 (1.96)	4.18 (1.75)	0.05 *
PA (min/day)	267 (40)	247 (39)	0.00 **
MVPA (min/day)	52.4 (13.6)	43.9 (13.6)	0.00 **
Sleep (hr/day)	10.64 (0.94)	10.86 (0.90)	0.03 *
ST (hr/day)	1.71 (1.18)	1.71 (1.15)	0.97
BMI	15.51 (1.49)	15.63 (1.61)	0.51

Note: PA = physical activity; MVPA = moderate-to-vigorous physical activity; BMI = body mass index; M = Mean; SD = standard deviation; * represents *p* < 0.05; ** represents *p* < 0.01.

**Table 3 jcm-11-05623-t003:** Meeting of 24-Hour Movement Guidelines among preschoolers’ boys and girls.

Movement Behaviors	Boys N (%)		Girls N (%)		*p*-Value
	Compliant (%)	Non-Compliant (%)	Compliant (%)	Non-Compliant (%)	
Not meeting guidelines	6 (3.3)	175 (96.7)	7 (4.97)	134 (95.03)	0.320
PA	53 (29.3)	128 (70.7)	11 (7.8)	130 (92.2)	0.000 **
SL	165 (91.2)	16 (8.8)	127 (90)	14 (10)	0.442
ST	99 (54.7)	82 (45.3)	73 (51.8)	68 (48.2)	0.341
PA_ST_SL	26 (14.36)	155 (85.63)	4 (2.84)	137 (97.16)	0.000 **
PA_ST	29 (16.02)	152 (83.98)	4 (2.84)	137 (97.16)	0.000 **
PA_SL	47 (25.97)	134 (74.03)	11 (7.80)	130 (92.19)	0.000 **
ST_SL	92 (50.82)	99 (49.17)	66 (46.81)	75 (53.19)	0.237

Note: PA = physical activity; SL = sleep; ST = screen time. ** represents *p* < 0.01.

**Table 4 jcm-11-05623-t004:** Associations between meeting 24-Hour Movement Guidelines met and fundamental motor skills in preschoolers.

	Locomotor Skills				Object Control Skills			
	β	SE	*p*	R^2^	β	SE	*p*	R^2^
PA	**0.48**	**0.06**	**0.00**		**0.21**	**0.05**	**0.00**	
SL	0.01	0.06	0.89	0.62	−0.03	0.05	0.56	0.49
ST	0.13	0.06	0.05		−0.04	0.05	0.47	
PA + ST + SL	0.12	0.21	0.55		0.12	0.17	0.47	
PA + ST	0.14	0.20	0.49	0.63	0.17	0.15	0.26	0.52
PA + SL	**0.31**	**0.09**	**0.001**		0.03	0.07	0.69	
ST + SL	**−0.16**	**0.07**	**0.02**		−0.06	0.05	0.28	

Note: PA = physical activity; SL = sleep; ST = Screen time. Bold represents significant relationship.

## Data Availability

The data presented in this study are available on request from the corresponding author. The data are not publicly available due to the lack of an online server for data storage.

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
