# Peer review of "Examining Relationships among Chinese Preschool Children’s Meeting 24-Hour Movement Guidelines and Fundamental Movement Skills"

_jcm, 2022, doi:10.3390/jcm11195623_

Round 1
Reviewer 1 Report
I would like to thank the authors for the great work done. The research is very well conducted and tables, figures and manuscript is carefully prepared. I really liked the idea in figure and how the data was presented. While the topic is important, time to time there are some points that are already very well studied, for example how boys are more active than girls. So this should be informed either from global perspective (meta-analysis) or from China.
In the introduction, FMS are discussed, but in the research project balance skills are not consider as a part of the study, why?
One which would be interesting is the background variables and associations to meeting the recommendations. Did you find out parents socioeconomic status, employment status or income and how would this be associated with meeting the recommendations?
Minor comments:
- Rewrite the last sentence in the abstract.
L 46 as an example. Please check your citations, there are [xx] marked with space before and without.
L 58 add space before 5.
L 67-68. Maybe at least shortly, discuss also about balance skills?
L 88 please check the numbers, 384-61 =322?
L 161 Maybe change to mins 1.71 hours? Same also in the next line.
L 231-233 How about reactivity? DO you consider that there are some reactivity, please discuss?
L 237 add space between , and for
L 237-238. I think that this is general result in several studies. Maybe you can even find a meta-analysis etc. that highlights this same issue, boys are more active than girls. If you are referring to one study only, then you should find a study from your country.
L 243 Significantly -> significantly
L 243 Where was this study (Tapia-Serrano), please add.
L 271-273 Please rewrite "is an idea" is not very academic way to write this sentence.
L 313-314 Why Sleep duration was not recorded via actigraphs? As this is nowadays possible? With questionnaires you can get information about the sleep duration, but not about the sleep quality (e.g. lots of disruptions).
L 380 Check WHO reference, it is incomplete
L 408 Q. -> Quar.
L 426 / L436 Basel can be deleted?
L 434 / 448 / 450 / 459 abbreviations are missing periods?
Author Response
Responses to reviewer 1
Thank you for your insightful comments in helping us to improve this research paper. We are excited to know that the importance of the paper has been acknowledged by our peer colleagues. In this revision, we have carefully addressed point-by-point each of your comments indicated below. It is our sincere hope that the revision would meet your standards. Thanks again for your professional review.
Responses to the comments from reviewer 1 report:
|
|
Comments |
Responses |
|
1 |
I would like to thank the authors for the great work done. The research is very well conducted and tables, figures and manuscript is carefully prepared. I really liked the idea in figure and how the data was presented. |
Thanks for acknowledging the quality of this paper and providing helpful comments to help us strengthen the quality of this manuscript. |
|
2 |
While the topic is important, time to time there are some points that are already very well studied, for example how boys are more active than girls. So this should be informed either from global perspective (meta-analysis) or from China. |
Thanks for your comment. We have provided literature on gender differences of children’s physical activity in this revision. See page 2 lines 68-70. Lines 260-264. |
|
3 |
In the introduction, FMS are discussed, but in the research project balance skills are not consider as a part of the study, why? |
Thanks for your comment. The balance skills are part of the FMS, but the instruments for assessing FMS (TGMD-3) we used in this study didn’t including testing items for balance. In addition, some researchers found that PA was not associated with balance skills. See page 2 line 73-76. Collectively, we didn’t include the balance skills in the present study. We will consider including this component in the future study. |
|
4 |
One which would be interesting is the background variables and associations to meeting the recommendations. Did you find out parents socioeconomic status, employment status or income and how would this be associated with meeting the recommendations? |
Thanks again for your suggestion. We have previously collected the data of SES, but did not add it to the original analysis. As suggested, we added SES as the covariate in the analysis. The corresponding changes were made in the Methods and Results sections. See page 4 lines151-156 and lines 229-230. |
|
5 Minor comments
|
5.1 Rewrite the last sentence in the abstract. |
We have rewritten the last sentence in the abstract. Please see lines 32-33. |
|
5.2 L 46 as an example. Please check your citations, there are [xx] marked with space before and without. |
We have made changes according to the comments. See line 46 and line 47 and so on. |
|
|
5.3 L 58 add space before 5. |
We have added space before 5. See line 59 |
|
|
5.4 L 67-68. Maybe at least shortly, discuss also about balance skills? |
Thanks for your comment. We have discussed the balance skills. See line 73-76. |
|
|
5.5 L 88 please check the numbers, 384-61 =322? |
Thanks for your suggestion. We have corrected the numbers in this version of paper. There were 62 preschoolers who did not provide complete data. |
|
|
5.6 L 161 Maybe change to mins 1.71 hours? Same also in the next line. |
Thanks for your comment. According the WHO’s 24-Hour movement guidelines, the time of ST and sleep are measured in hours, and PA time is measured in minutes. We decided to stick to the unit of the guidelines. Thus, we did not change to mins 1.71 hours. |
|
|
5.7 L 231-233 How about reactivity? DO you consider that there are some reactivity, please discuss? |
Thanks for your comment. We have discussed and modified the reactivity according to the comments. See line 251-255. |
|
|
5.8 L 237 add space between , and for |
We have added space between, and for. See line 260 |
|
|
5.9 L 237-238. I think that this is general result in several studies. Maybe you can even find a meta-analysis etc. that highlights this same issue, boys are more active than girls. If you are referring to one study only, then you should find a study from your country. |
Thanks for your comment. We have provided literature on gender differences of children’s physical activity in this revision. See page 2 lines 68-70. Lines 261-265. |
|
|
5.10 L 243 Significantly -> significantly |
We have changed Significantly to significantly. See line 269. |
|
|
5.11 L 243 Where was this study (Tapia-Serrano), please add. |
The reference [36] is the study (Tapia-Serrano). See line 469 |
|
|
5.12 L 271-273 Please rewrite "is an idea" is not very academic way to write this sentence. |
We have rewritten this sentence. See line 301-303. |
|
|
5.13 L 313-314 Why Sleep duration was not recorded via actigraphs? As this is nowadays possible? With questionnaires you can get information about the sleep duration, but not about the sleep quality (e.g. lots of disruptions). |
Thanks again for your comment. It is possible to track sleep duration be the ActiGragh nowadays, but we encountered some difficulties while using the accelerometers to objectively measure sleep duration. A major challenge is that some children thought the ActiGragh accelerometers would cause discomfort if they wore the accelerometers on the right hip during sleep in the study. As a result, we recorded the sleep duration by questionnaires. We added it as a limitation of this study in the Discussion section. We will consider using the more objective measurement to record the sleep duration in the future study. See lines 351-356. |
|
|
5.14 L 380 Check WHO reference, it is incomplete |
We have checked and made corrections based on your suggestion. See lines 417. |
|
|
5.15 L 408 Q. -> Quar. |
Thanks again for your comment. We checked many times and the abbreviation is this expression (Q) instead of Quar. |
|
|
5.16 L 426 / L436 Basel can be deleted? |
We have made corrections based on your suggestion. See line 469,484. |
|
|
5.17 L 434 / 448 / 450 / 459 abbreviations are missing periods? |
We have made corrections based on your suggestion. See line 483/497/499/508. |
Reviewer 2 Report
In the study, it was aimed to examine the status of meeting the 24-Hour Movement Guidelines and fundamental movement skills (FMS) and the differences between the genders of Chinese preschool children.
Although the study is a descriptive and local , it was evaluated that it would contribute to the scientific literature, when considered the limited data in this age group.
However, the following corrections and clarifications are suggested.
It is said that FMS scoring is done in the material and method section. What was this made for? How are these scores expressed? Are the data numeric or not? How was objectivity achieved during the process of obtaining this data? It should be written more clearly.
RMSEA, CFI, TLI abbreviations are mentioned in the section of findings. What do they mean. Explanations of these abbreviations should be written.
One of the most basic findings of the study is the differences between the gender. Researchers need to further discuss the reasons for these differences. It is not enough to explain this only with cultural and traditional differences. Cultural and traditional influence explain the differences between countries.
Author Response
Responses to reviewer 2
Thank you for your insightful comments in helping us to improve this research paper. We are excited to know that the importance of the paper has been acknowledged by our peer colleagues. In this revision, we have carefully addressed point-by-point each of your comments indicated below. It is our sincere hope that the revision would meet your standards. Thanks again for your professional review.
Responses to the comments from reviewer 2 report:
|
|
Comments |
Responses |
|
1 |
In the study, it was aimed to examine the status of meeting the 24-Hour Movement Guidelines and fundamental movement skills (FMS) and the differences between the genders of Chinese preschool children. Although the study is a descriptive and local, it was evaluated that it would contribute to the scientific literature, when considered the limited data in this age group.
|
Thanks for acknowledging the quality of this paper and providing helpful comments to help us strengthen the quality of this manuscript. |
|
2 |
However, the following corrections and clarifications are suggested. It is said that FMS scoring is done in the material and method section. What was this made for? How are these scores expressed? Are the data numeric or not? How was objectivity achieved during the process of obtaining this data? It should be written more clearly.
|
Thanks again for your suggestion. We have supplemented and explained the motor skills scoring criteria. See lines 136-142.
|
|
3 |
RMSEA, CFI, TLI abbreviations are mentioned in the section of findings. What do they mean. Explanations of these abbreviations should be written.
|
Thanks again for your suggestion, we have supplemented the mean of RMSEA, CFI, TLI. See lines 209-210.
|
|
4 |
One of the most basic findings of the study is the differences between the gender. Researchers need to further discuss the reasons for these differences. It is not enough to explain this only with cultural and traditional differences. Cultural and traditional influence explain the differences between countries.
|
Thanks for your comment. We have provided literature on gender differences of children’s physical activity in this revision. See page 2 lines 68-70. Lines 260-264. |